# Immunohistochemistry of the Gut-Associated Lymphoid Tissue (GALT) in African Bonytongue (*Heterotis niloticus,* Cuvier 1829)

**DOI:** 10.3390/ijms24032316

**Published:** 2023-01-24

**Authors:** Eugenia Rita Lauriano, Alessio Alesci, Marialuisa Aragona, Simona Pergolizzi, Anthea Miller, Kristina Zuwala, Michal Kuciel, Giacomo Zaccone, Antonino Germanà, Maria Cristina Guerrera

**Affiliations:** 1Department of Chemical, Biological, Pharmaceutical, and Environmental Sciences, University of Messina, 98166 Messina, Italy; 2Zebrafish Neuromorphology Lab, Department of Veterinary Sciences, University of Messina, 98168 Messina, Italy; 3Department of Veterinary Sciences, University of Messina, Polo Universitario dell’Annunziata, 98168 Messina, Italy; 4Department of Comparative Anatomy, Institute of Zoology and Biomedical Researches, Faculty of Biology, Jagiellonian University in Krakow, 30-501 Krakow, Poland; 5Poison Information Centre, Department of Toxicology and Environmental Disease, Faculty of Medicine, Jagellonian University, Kopernika 15, 30-501 Krakòw, Poland

**Keywords:** *Heterotis niloticus*, GALT, lymphoid tissue, phylogenesis

## Abstract

*Heterotis niloticus* is a basal teleost, belonging to the Osteoglossidae family, which is widespread in many parts of Africa. The digestive tract of *H. niloticus* presents similar characteristics to those of higher vertebrates, exhibiting a gizzard-like stomach and lymphoid aggregates in the intestinal lamina propria. The adaptive immune system of teleost fish is linked with each of their mucosal body surfaces. In fish, the gut-associated lymphoid tissue (GALT) is generally a diffuse immune system that represents an important line of defense against those pathogens inhabiting the external environment that can enter through food. The GALT comprises intraepithelial lymphocytes, which reside in the epithelial layer, and lamina propria leukocytes, which consist of lymphocytes, macrophages, granulocytes, and dendritic-like cells. This study aims to characterize, for the first time, the leukocytes present in the GALT of *H. niloticus,* by confocal immuno- fluorescence techniques, using specific antibodies: toll-like receptor 2, major histocompatibility complex class II, S100 protein, serotonin, CD4, langerin, and inducible nitric oxide synthetase. Our results show massive aggregates of immune cells in the thickness of the submucosa, arranged in circumscribed oval-shaped structures that are morphologically similar to the isolated lymphoid follicles present in birds and mammals, thus expanding our knowledge about the intestinal immunity shown by this fish.

## 1. Introduction

*Heterotis niloticus* (Cuvier, 1829), commonly known as the African bonytongue, is a species of ray-finned fish belonging to the Arapaimidae family and is the only species in the genus *Heterotis.* This fish is native to many countries of Africa and because of its good meat quality, with a high protein content, it holds high commercial value for many Nigerians [1,2]. Numerous studies concern the various aspects of the reproduction, feeding, biology, and ecology of this fish [1]; furthermore, its basal position in the phylogeny as the Osteoglossiforms makes this fish interesting for studying evolutionary processes [3,4]. In a previous study, Guerrera et al. [4] described the anatomy and morphology of the African bonytongue’s digestive system, which presents similarities with reptiles and birds. This fish has a gizzard-like stomach that is adapted to chopping and shredding food [5]; it is a bilobed organ that is divided into a *pars glandularis* and a thick-walled *pars muscularis*. *H. niloticus* is an omnivorous fish; its diet consists of a wide variety of bottom-dwelling food sources, such as insect larvae, microcrustaceans, and hard seeds. The thick-walled gizzard, which contains sand, aids in the digestion of seed coats [2,6]. The gizzard continues in the form of the foregut and two blind pyloric appendages, which perform specific functions, including immune defense against the presence of mucosa-associated lymphoid tissues (MALT). In *H. niloticus*, as in other fish, the intestinal posterior segment is also immunologically active [7]. Teleost fish possess an adaptive immune system that is aggregated with each of their mucosal body surfaces. The main mucosa-associated lymphoid tissues (MALT) of teleosts are the skin-associated lymphoid tissue (SALT), which contains diffuse lymphoid tissue and microbiota [8,9], the gill-associated lymphoid tissue (GIALT), and the intrabronchial lymphoid tissue (ILT) [10], the recently discovered nasopharynx-associated lymphoid tissue (NALT), which is located in the olfactory organ [8,11], and, finally, the buccal-pharyngeal-associated lymphoid tissues (OFALT) [12]. The best-described MALT in teleosts is the gut-associated lymphoid tissue (GALT), which plays an important role in fish health [13,14,15,16,17]. Generally, the GALT has a similar morphology in the various species of fish, although there may be structural differences regarding the gut and the related GALT among herbivorous, carnivorous, and omnivorous fish [18]. Teleost fish are the earliest living organisms to possess the most important components of an adaptive immune system, such as the major histocompatibility complex (classes I and II) and the B and T cells [16,19].

The adaptive mucosal immune responses in teleost fish have been investigated for many years in studies conducted in rainbow trout (*Oncorhyncus mykiss*) and plaice (*Pleuronectens platessa*), concerning oral and parenteral immunization [4,5,8].

The fish’s GALT comprises different types of leukocytes: intraepithelial lymphocytes (IELs) and lamina propria leukocytes (LPLs), such as lymphocytes and phagocytic cells (granulocytes, macrophages, and dendritic-like cells) [13]. 

The lamina propria and intestinal epithelium are separated by a thin basement membrane and form two different immunological regions [16,20]. In some teleost species, epithelium-associated and mucosal macrophages have been reported [21], along with epithelial macrophages engulfing apoptotic epithelial cells and potentially harmful microbes; the mucosal macrophages function as antigen-presenting cells and cytokine producers. Furthermore, these macrophages can be highly innervated, performing a neuroprotective role in the teleost enteric nervous system [22,23].

In the teleost, both B and T lymphocytes have been characterized in the intestinal epithelium and in the lamina propria [8]. The clusters of B cells and IgM immunoglobulins produced by these lymphocytes represent the first line of defense against those pathogens that are introduced to the body along with food [16,24]. Numerous studies have documented the presence of diffusely organized GALT in the gut mucosa of teleosts; however, there are few reports about the existence of complex lymphoid tissue in the form of Peyer’s patches in fish [24,25]. Lymphoid aggregates have been observed along the intestines of amphibians [26]; recently, in African lungfish, intestinal mucosa-encapsulated and -unencapsulated lymphoid aggregates were described [26,27]. In this study, we have investigated, for the first time, the immunity features of *H. niloticus* GALT, using antibodies directed against the toll-like receptor (TLR) 2, major histocompatibility (complex class II (MHCII)), S100 protein, serotonin (5-hydroxytryptamine; 5-HT), CD4, langerin/CD207, and inducible nitric oxide synthetase (iNOS). In previous studies, we have used these antibodies to characterize immune cells, such as lymphocytes, macrophages, dendritic-like cells, and mast cells (MC) in the fish’s different tissues and organs. TLR2 is present and highly conserved across all vertebrate species, being expressed in immune and non-immune cell types [28,29,30,31,32,33,34,35,36,37]. S100 is commonly used as a marker for macrophages [38], Langerhans cells [39,40], and MCs [41,42,43]. In addition, 5-HT is a neurotransmitter expressed in the central nervous system and gastrointestinal tract, which is presumably conserved in all vertebrate species [44]; it is involved in the activation of T and natural killer (NK) cells, along with the production of chemotactic factors via macrophages [45]. iNOS peptides are involved in the function of all types of vertebrate immune cells [46]. Langerin/CD207 is a specific marker of Langerhans cells. This antibody was used to identify Langerhans-like cells in the spleen, kidney, and gut of several bony fish species [29,47,48,49]. Soleto and colleagues [50] have identified DCs in the intestine of rainbow trout, using CD8α+ and MHC II [50,51]. CD4 molecules have been reported in lower vertebrates, such as teleosts. The CD4^+^ helper T cells in fish are similar to those present in the higher vertebrates. CD4 is a membrane glycoprotein that functions as a co-receptor during immune recognition between the TCR and the MHC II/peptide complex [8,52]. 

The aim of our study was to deepen scientific knowledge regarding the morphology and structure of the GALT in this basal species, thus adding one more piece to the fascinating, although complicated, phylogenetic evolution of immune tissues in vertebrates.

## 2. Results

The examination of both transverse and longitudinal sections of the pyloric ceca and intestine terminal region (rectum) reveals massive aggregates of immune cells. GALT was seen in two different shapes: the immune cells present in the thickness of the submucosa and the lamina propria are arranged in circumscribed oval-shaped structures that are morphologically similar to the isolated lymphoid follicles (ILFs) present in birds and mammals. Furthermore, scattered or clustered immune cells are densely packed in the lamina propria and submucosa. The ILFs consist of non-encapsulated, dense clusters of lymphocytes of various sizes and macrophages. Double immunolabeling with antibodies against TLR2 and MHCII reveals positive DC-like cells in the aggregates of lymphoid tissue (Figure 1); these cells are strongly positive with TLR2 and Langerin/CD 207 (Figure 2). A large accumulation of lymphocytic cells that are marked with CD4 and colocalized with serotonin can be seen, especially in the cells located at the periphery of the lymphoid structure (Figure 3). Numerous S100 and CD4-positive cells are scattered under the epithelium and there is no colocalization with serotonin, which marks the neuroendocrine cells in the epithelium (Figure 4 and Figure 5). Strong colocalization between TLR2 and iNOS is evident in the submucosa cells (Figure 6).

Quantitative analysis revealed an equal number of cells that were immunopositive for each antibody tested (Table 1).

## 3. Discussion

*H. niloticus* is particularly interesting in the study of the phylogenesis of vertebrates because, although it presents primitive characteristics, it has anatomical specializations that are similar to those of the higher vertebrates. Some of the primitive features belonging to *H. niloticus* are an elongated and robust body, dorsal and anal fins that are elongated and posteriorly positioned, a rounded caudal fin, and strong, large scales. Furthermore, *H. niloticus* presents reduced lamellar surfaces and a large gas bladder that helps them to acquire O_2_ from the environment [53]. On the other hand, it presents a stomach consisting of a proventriculus and a ventriculus (gizzard), found in birds and reptiles, as an organ of digestion, due to its omnivorous feeding [2]. Moreover, the wall of the alimentary tract of *H. niloticus* shows a strong similarity to that of the higher vertebrates. It is composed of four layers, which, proceeding from the inside outward, comprise the mucosa, submucosa, inner circular layer, outer longitudinal layer of muscularis, and serosa [4,54]. In agreement with Guerrera et al. [4], in this study, massive aggregates of lymphoid and innate immune cells were observed in the intestinal mucosa and submucosa, constituting structures that were arranged in circumscribed oval-shaped forms resembling mammalian ILFs. Lymphoid aggregates have been found in all vertebrates, including amphibians, reptiles, and birds [55,56,57,58,59,60]. GALT is the main MALT of teleosts, being continuously exposed to a pathogen-rich environment in either freshwater or seawater. In teleosts, the highly organized lymphoid organs, such as mesenteric lymph nodes and Peyer’s patches, which have been observed in birds and mammals, are missing. However, in some studies, the existence of ILT [10,14,16,61,62], and the bursa in Atlantic salmon (*Salmo salar,* Linnaeus 1758), which are an analog of the avian bursa of Fabricius [63], were reported as the organized lymphoid structure of teleosts. In the phylogenesis of vertebrates, the lymphoid system has increased its complexity and structural organization, to become increasingly efficient and specialized. Intestinal lymphoid aggregates lack germinal centers, as described in the lamina propria of birds; those found in cold-blooded vertebrates may represent primitive versions of the cryptopatches that are present in the mammalian intestine [26,64]. Teleosts can present B and T lymphocytes in separate areas of the MALT of the gut, gills, and nasopharynges, as with those amphibians and reptiles that separate the B and T cell areas in the spleen [7]; finally, birds and mammals present lymphoid organs that are highly organized, with separated B and T cell areas and well-demarcated germinal centers [7,65]. Several studies have described the fish GALT as consisting of immune cells (lymphocytes, macrophages, dendritic cells, and plasma cells), which are presented in clusters along the mucosa of the alimentary canal and the intraepithelial lymphocytes distributed among the enterocytes [8,24,66]. The gut-associated lymphoid tissue of *H. niloticus* seems to be present in two different shapes. Our results showed scattered clusters of immune cells in the mucosal lamina propria of the pyloric ceca and rectum (the terminal part of the hindgut), using confocal immunofluorescence. The gut mucosal epithelia play a significant role in fish immunology [29,67]. Using confocal microscopy, we have immunohistochemically characterized the DC-like cells with antibodies against TLR2 and MHCII; the presence of these cells was confirmed by colocalization between TLR2 and Langerin/CD 207; furthermore, the S100 protein and iNOS have been used to mark the macrophages and CD4 antibodies to characterize the T lymphocytes. CD4+ helper T cells can be found in the teleost’s gastrointestinal lamina propria [13,68,69]. These cells express CD4 on their surface for specific antigen recognition. Toll-like receptors are involved in the recognition of pathogens, through specialized antigen-presenting cells (APCs). These antigen-presenting cells include macrophages, granulocytes, and dendritic-like cells, as well as B cells. The presentation of the antigen takes place via MHCII, activating T cells that proliferate and produce inflammatory cytokines [70]. The distribution of these innate and adaptive immune cells, as delineated by the pattern of anti-TLR2, anti-MHCII, anti-Langerin/CD 207, anti-iNOS, and anti-CD4 recognition, was similar to that documented in other studies [8,23,51]. Recent findings have shown rainbow trout to be a model for intestinal immune responses. In the intestine lamina propria, pathogens can be taken up by the antigen-presenting cells (APCs) and then presented to CD4-T cells. Consequently, the B lymphocytes, activated by the cytokines produced by T cells, proliferate and differentiate into plasma cell-like cells, resulting in the production of antibodies [16,71].

This interaction between the immune cells in fish is very similar to that found in mammals; thus, improving this knowledge could be useful for formulating new vaccines and finding new models by which to improve mammalian mucosal immunology. One very interesting finding is the positivity to 5-HT; the endocrine cells of the intestinal epithelium are positive for this neurotransmitter. Moreover, the lymphocytes that are localized in the peripheral zone of the lymphoid structures present colocalization between CD4 and serotonin. The immune cells express serotonin receptors and serotonin transporter (SERT), which are known serotonergic components of the immune cells [72,73,74]. T cells take up serotonin via SERT and then express numerous 5-HT receptors that are involved in the proliferation of T lymphocytes [74,75]. 

## 4. Materials and Methods

The paraffin-embedded tissue of males and females of the adult African bonytongue, *H. niloticus* (Cuvier, 1829), from a previous study was used for this research. In that study, the fish’s digestive system (from tongue to anus) was sampled and processed for routine histological study (for details, see Guerrera et al. [4]).

### 4.1. Immunofluorescence 

To identify the localization of anti-TLR2, serotonin (5-HT), MHCII, Langerin/H-4, the S100 protein, and CD4 antibodies, an immunohistochemistry investigation was carried out on the pyloric ceca and intestine’s terminal region (rectum). Deparaffinized and rehydrated serial slices (10 µm thick) were rinsed in Tris-HCl solution (0.05 M, pH 7.5) with 0.1% bovine serum albumin and 0.2% Triton-X 100. Slices were incubated in a 0.3% H2O2 (PBS) solution to prevent endogenous peroxidase activity; finally, fetal bovine serum (F7524, Sigma-Aldrich, St. Louis, Missouri, USA) was added to the washed sections for 30 min to prevent nonspecific binding, then the primary antibodies were incubated. Anti-TLR2 and anti-serotonin (5-HT) polyclonal antibodies were used in the double-label experiments, with a monoclonal antibody for MHCII, Langerin/H-4, S100 protein, and CD4 (for details, see Table 2). A humid chamber was used for overnight incubation at 4°C. Subsequently, the sections were rinsed in buffer and incubated for 40 min at room temperature with Alexa Fluor IgG (H + L) secondary antibodies (for details, see Table 2) in a dark, humid chamber. Finally, Fluoromount Aqueous Mounting Medium was applied to mount the dehydrated sections (Sigma-Aldrich, Burlington, MA, USA).

### 4.2. Laser confocal immunofluorescence

Sections were analyzed and images were acquired using a Zeiss LSMDUO confocal laser scanning microscope with a META module (Carl Zeiss MicroImaging GmbH, Germany) microscope, the LSM700 AxioObserver. The Zen 2011 (LSM 700, Zeiss software, Oberkochen, Germany) built-in “colocalization view” was used to highlight the expression of both antibodies’ signals [76,77,78], in order to produce a “colocalization” signal, along with the display profile, the scatter plot and fluorescent signal measurements. Each image was rapidly acquired to minimize photodegradation.

### 4.3. Statistical Analysis

Data were gathered for quantitative analysis through the examination of ten sections and twenty fields per sample. The cell positivity was assessed using ImageJ software 1.53e. The plugin, “Analyze particles”, was used to count the number of cells. We enumerated the number of cells in each field that were positive for TLR2, Langerin/CD207, 5-HT, MHC II, S100, and CD4, using a SigmaPlot, version 14.0 (Systat Software, San Jose, CA, USA). A one-way ANOVA and Student’s *t*-test were used to assess the normally distributed data. Data means and standard deviations (SD) are shown as ** *p* 0.01 and * *p* 0.05. 

## 5. Conclusions

For the first time in this study, immune cells of the GALT of *H. niloticus* were characterized, confirming the presence of organized lymphoid structures similar to those seen in the higher vertebrates. Further studies could be useful to better understand the phylogeny of the vertebrate immune system and to consider the possibility of vaccination strategies for highly commercial species, such as *H. niloticus*.

## Figures and Tables

**Figure 1 ijms-24-02316-f001:**
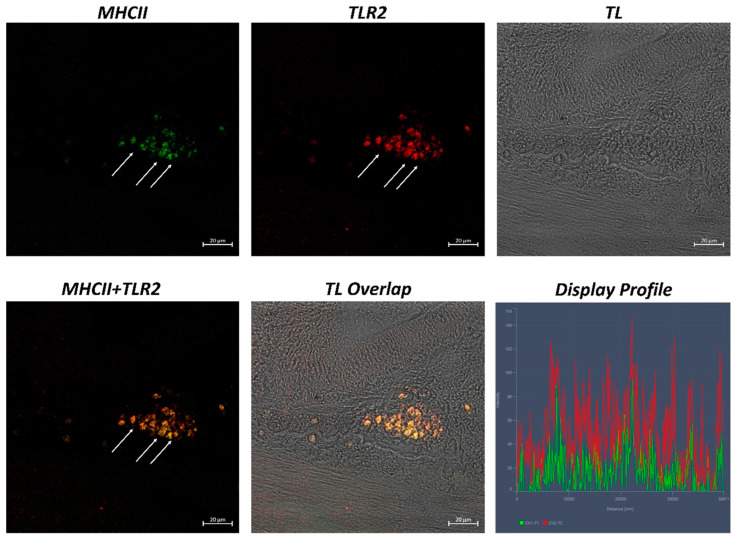
Section of the *H. niloticus* gut (immunofluorescence 40×, scale bar 20 μm). Macrophages that are positive for TLR2 and MHC II can be noted in the lymphoid tissue aggregates (marked by arrows). The “Display profile” function highlights the colocalization. TL = transmitted light.

**Figure 2 ijms-24-02316-f002:**
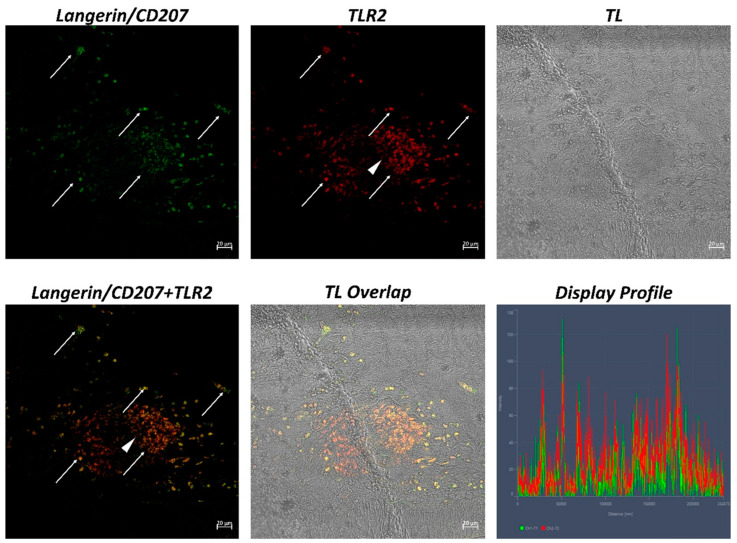
Section of the *H. niloticus* gut (immunofluorescence 20×, scale bar 20 μm). DC-like cells that are immunoreactive for Langerin and TLR2, organized in clusters, can be seen (arrows). Some immune cells are positive exclusively for TLR2 (see the arrowhead). The “Display profile” function highlights the colocalization. TL = transmitted light.

**Figure 3 ijms-24-02316-f003:**
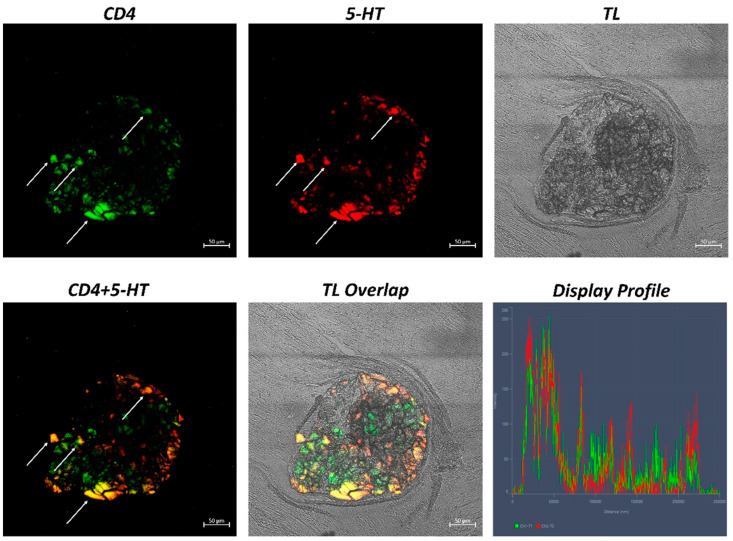
Section of the *H. niloticus* gut (immunofluorescence, 40×, scale bar 50 μm. The presence of an important accumulation of densely packed lymphocytes is highlighted by CD4 and 5-HT positivity (arrows). The “Display profile” function highlights the colocalization. TL = transmitted light.

**Figure 4 ijms-24-02316-f004:**
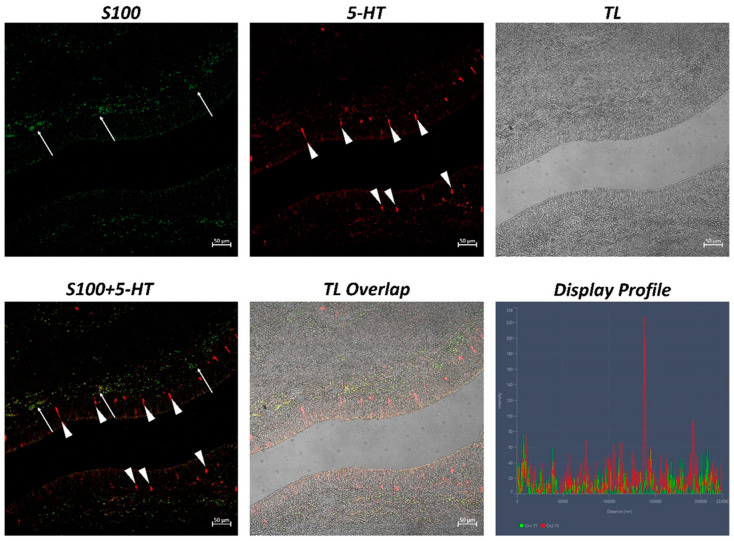
Section of the *H. niloticus* gut (immunofluorescence 20×, scale bar 50 μm). Immune cells that are positive for S100 are evident in the submucosa (arrows), while the neuroendocrine cells (5-HT positive) are evident in the epithelium (arrowheads). The “Display profile” function highlights the absence of colocalization. TL = transmitted light.

**Figure 5 ijms-24-02316-f005:**
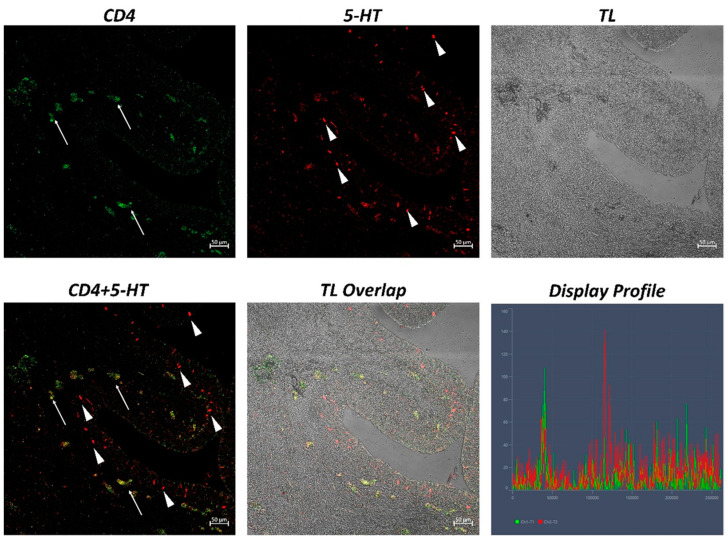
Section of the *H. niloticus* gut (immunofluorescence 20×, scale bar 50 μm). Lymphocytes that are positive for CD4 and 5-HT are scattered in the submucosa (arrows). The presence of neuroendocrine cells that are immunoreactive to 5-HT can also be noted (arrowheads). The “Display profile” function highlights the colocalization. TL = transmitted light.

**Figure 6 ijms-24-02316-f006:**
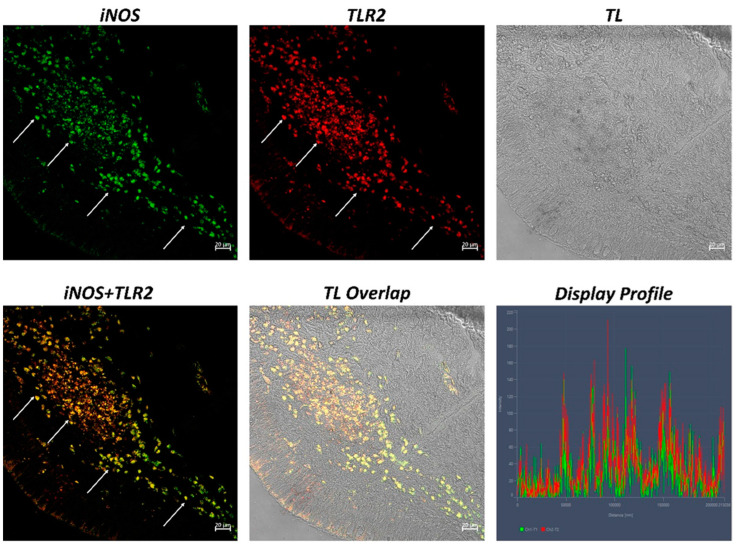
Section of the *H. niloticus* gut * immunofluorescence (* 40×, scale bar 20 μm). Numerous immune cells, which are densely organized, colocalize for iNOS and TLR2 (arrows) in the submucosa. The “Display profile” function highlights the colocalization. TL = transmitted light.

**Table 1 ijms-24-02316-t001:** Statistical analysis data. (Mean values ± standard deviation) (*n* = 3).

	No. of Positive Immune Cells
TLR-2	476.38 ± 45.29 *
MHC class II	447.44 ± 38.20 **
Langerin	329.46 ± 28.01 *
CD4	365.25 ± 33.06 *
5-HT	313.94 ± 29.36 *
S100	406.36 ± 39.85 *
iNOS	418.42 ± 47.31 **
TLR2 + MHC II	416.03 ± 37.78 *
TLR2 + Langerin	301.53 ± 30.25 **
CD4 + 5-HT	307.48 ± 20.34 *
TLR2 + iNOS	403.94 ± 32.75 **

* *p* ≤ 0.05; ** *p* ≤ 0.01.

**Table 2 ijms-24-02316-t002:** Antibodies data.

**Primary Antibodies**	**Supplier**	**Catalog Number**	**Source**	**Dilution**	**Antibody ID**
TLR-2 (pAb)	Active Motif	40981	Rabbit	1:125	AB_2750977
Anti-serotonin (5HT)	Sigma-Aldrich	S5545	Rabbit	1:300	AB_477522
Langerin/H-4	Santa Cruz Biotechnology	sc-271272	Mouse	1:250	AB_10611518
MHC class II (Y-Ae)	Santa Cruz Biotechnology	Sc-32247	Mouse	1:250	AB_627939
S100 (s161)	Santa Cruz Biotechnology	Sc-53438	Mouse	1:100	AB_630214
CD4 MT310	Santa Cruz Biotechnology	Sc-19641	Mouse	1:100	AB_627055
iNOS	Santa Cruz Biotechnology	Sc7271	Mouse	1:200	
**Secondary Antibodies**	**Supplier**		**Source**	**Dilution**	**Antibody ID**
Alexa Fluor 488 anti-mouse IgG FITC conjugated	Invitrogen	A-21202	Donkey	1:300	AB_141607
Alexa Fluor 594 anti-rabbit IgG TRITC conjugated	Invitrogen	A32754	Donkey	1:300	AB_2762827

## Data Availability

All data presented in this study are available from the corresponding author, upon responsible request.

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
