# Peer review of "Immunohistochemistry of the Gut-Associated Lymphoid Tissue (GALT) in African Bonytongue (Heterotis niloticus, Cuvier 1829)"

_ijms, 2023, doi:10.3390/ijms24032316_

Round 1

Reviewer 1 Report

 The article deals with the study of the immune system in teleost fish, locating in the intestinal tract several compounds that are supposed, in homology with other vertebrates, to be related to this system in fish.

This basic study addresses a very interesting and promising topic. However, a more in-depth bibliographic review is needed as the article lacks specific information on fish, the introduction is very general and should be adapted to what is really known in teleosts.

As for the results, the text in the figures legends should indicate the tissue shown.

In addition, the resolution of the images is not good and should be improved and/or images of higher quality should be attached.

The discussion seems more like an extended introduction and results section than a real discussion, where the results obtained and future perspectives are explained in more depth.

5HT is not a peptide!

In my opinion this article needs to be revised in depth and made more scientifically robust.

Reviewer 2 Report

I have only one question to the authors

All antibodies used in the present study were sourced from rabbits, mouse, and donkey. These sources are too far away from the fish sources. Even though antibodies differ within different fish species. How the authors confirm the antigen-antibody reaction with antibodies prepared from sources other than fish species?

Minor revisions

Use the abbreviated name of this fish species “H. niloticus” after its first appearance in the text.

Line 40: Add suitable reference for this information.

Line 97: Authors should write the main aims of their study.

Line 206: What did the authors mean with “previous studies”? – Add related references

References - Minor revisions in the journal names and writing of Latin names

Line 326: Monopterus albus (italic)

Line 379: Ecological Indicators

Line 410: Heterotis niloticus (italic)

Line 418: The Journal of Immunology

Line 423: Frontiers in Immunology

Line 449: Frontiers in Immunology

Line 463: Heteropneustes fossilis (italic)

Round 2

Reviewer 1 Report

The authors have made the necessary changes in order to accept the work. Although I have not received their replies to my considerations, could you please send them to me? 

Reviewer 2 Report

Authors have properly addressed the comments raised by the anonymous reviewer and merits acceptance